# Iodine Intake in Norwegian Women and Men: The Population-Based Tromsø Study 2015–2016

**DOI:** 10.3390/nu12113246

**Published:** 2020-10-23

**Authors:** Ahmed A Madar, Espen Heen, Laila A Hopstock, Monica H Carlsen, Haakon E Meyer

**Affiliations:** 1Department of Community Medicine and Global Health, Institute of Health and Society, University of Oslo, 0318 Oslo, Norway; e.k.heen@medisin.uio.no (E.H.); h.e.meyer@medisin.uio.no (H.E.M.); 2Department of Community Medicine, Faculty of Health Sciences, UiT The Arctic University of Norway, 9037 Tromsø, Norway; laila.hopstock@uit.no; 3Department of Nutrition, Institute of Basic Medical Sciences, University of Oslo, 0318 Oslo, Norway; m.h.carlsen@medisin.uio.no; 4Division of Mental and Physical Health, Norwegian Institute of Public Health, 0213 Oslo, Norway

**Keywords:** 24-h iodine, urinary iodine excretion, food frequency questionnaire, population-based studies, adult, iodine intake

## Abstract

Ensuring sufficient iodine intake is a public health priority, but we lack knowledge about the status of iodine in a nationally representative population in Norway. We aimed to assess the current iodine status and intake in a Norwegian adult population. In the population-based Tromsø Study 2015–2016, 493 women and men aged 40–69 years collected 24-h urine samples and 450 participants also completed a food frequency questionnaire (FFQ). The 24-h urinary iodine concentration (UIC) was analyzed using the Sandell–Kolthoff reaction on microplates followed by colorimetric measurement. Iodine intake was estimated from the FFQ using a food and nutrient calculation system at the University of Oslo. The mean urine volume in 24 h was 1.74 L. The median daily iodine intake estimated (UIE) from 24-h UIC was 159 µg/day (133 and 174 µg/day in women and men). The median daily iodine intake estimated from FFQ was 281 µg/day (263 and 318 µg/day in women and men, respectively). Iodine intake estimated from 24-h UIC and FFQ were moderately correlated (Spearman rank correlation coefficient r = 0.39, *p* < 0.01). The consumption of milk and milk products, fish and fish products, and eggs were positively associated with estimated iodine intake from FFQ. In conclusion, this shows that iodine intake estimated from 24-h UIC describes a mildly iodine deficient female population, while the male population is iodine sufficient. Concurrent use of an extensive FFQ describes both sexes as iodine sufficient. Further studies, applying a dietary assessment method validated for estimating iodine intake and repeated individual urine collections, are required to determine the habitual iodine intake in this population.

## 1. Introduction

Iodine is an essential trace element and it is required for the production of thyroid hormones (thyroxine and triiodothyronine) which are essential for regulation of energy metabolism in adults [1]. Too little and too much iodine can both increase the risk of thyroid dysfunction, and iodine deficiency can lead to a variety of health consequences known as iodine deficiency disorders (IDDs) [2]. Globally, before 1990, only a few countries were iodine sufficient, but comprehensive progress has been made since the primary intervention strategy for IDD control, i.e., universal salt iodization was adopted in 1993 [3,4]. Despite this global progress, recently, Europe has been one of the regions in the world where mild iodine deficiency has been most prevalent [5,6,7,8,9].

In Norway, IDD used to be endemic, but fortification of animal fodder since the early 1950s (2 mg of iodine per kilogram) contributed to the elimination of goiter and the prevention of IDD due to consumption of milk and dairy products [10]. Since then, milk, dairy products, fish, and fish products have been the main sources of iodine in the diet of Norwegians, contributing about 80% of the total iodine intake [11,12]. Although iodized salt is an important source of iodine in many countries, iodine fortification is only permitted in table salt (five milligrams of iodine per kilogram) in Norway. Furthermore, in recent years, consumption of milk and seafood has decreased [13]. A reduction of iodine content in milk has also been reported, linked to changed composition of cow fodder and possibly less use of iodophors for cleaning milk teats [14].

For many years, Norway has been considered to be an iodine-sufficient country, but recent findings have confirmed inadequate iodine intake among subgroups including pregnant and lactating women, vegans, and certain immigrant groups [9,15,16,17].

The World Health Organization (WHO) recommends regular mapping of iodine status in the population to counteract iodine deficiency, and the Norwegian Nutrition Council has recently recommended systematic monitoring of iodine status in the Norwegian population [1,18].

The recommended daily intake (RDI) of iodine in the Nordic countries (NNR12) is 150 μg/day for adults [18], and the estimated average requirement (EAR) is 100 μg/day, in line with the WHO guideline [19]. Around 90% of iodine intake is believed to be excreted via urine within 24 h, and the median urinary iodine concentration (UIC) is currently the most practical method for assessing the overall iodine status in a population [20]. However, UIC cannot be used to quantify the proportion of individuals with iodine deficiency or iodine excess. Calculating iodine intake from diet and dietary iodine supplements might provide a more complete understanding of habitual (long-term) iodine intake in the population. The WHO recommends a median UIC of at least 100 μg/L to prevent IDD in the general population [1]. In addition, no more than 20% of the UIC samples should be lower than 50 µg/L and a median UIC higher than 600 µg/L is defined as ‘excessive iodine intake [21].

For ensuring adequate iodine intake in all groups of the population, the Norwegian Nutrition Council has recently recommended universal salt iodization [18]. However, knowing that iodine deficiency and excess iodine both have adverse health consequences, it is important to monitor the iodine status in the population.

By drawing from the population-based Tromsø Study, the aim of the present study was to assess the current iodine status and intake in a Norwegian adult population.

## 2. Materials and Methods

The Tromsø Study is a population based, prospective, multipurpose study consisting of seven repeated surveys conducted from 1974 to 2016 (Tromsø 1–Tromsø 7), inviting whole birth cohorts and random samples of the inhabitants in the Tromsø Municipality, Northern Norway. Data collection included questionnaires and interviews, biological sampling, and clinical examinations. Tromsø is situated ~400 km north of the Arctic Circle, and has approximately 76,000 inhabitants. The seventh survey (Tromsø 7) was carried out during 2015–2016 and all 32,591 inhabitants aged 40 years and above were invited. A total of 21,083 women and men participated (65%) [22]. After excluding participants with self-reported heart failure, stroke, liver disease, participants who had started treatment with diuretics during the preceding two weeks, or with conditions making it difficult to collect urine (e.g., poor general health condition), a random subsample of 608 participants aged 40–69 years was invited for a 24-h urine collection substudy. Of these, 496 (82%) responded and collected urine. There were no statistically significant differences in body mass index (BMI) or education level between those collecting and not collecting urine. Three participants with undetermined urine volume were excluded from the analyses. Thus, a total of 493 participants were included in urine analyses.

### 2.1. Data Collection

#### 2.1.1. Twenty-Four-Hour Urine Collection

As previously reported [23], the participants were instructed (oral and written) to collect all urine during a 24-h period and record the first and last urination time points, as well as any missed voids or other irregularities. The participants should void their bladder in the first morning of collection and discard this urine, and then collect all urine until and including the morning void of the day after. After returning the 24-h urine specimens, the containers were well stirred, and the volume was read and recorded. Urine samples were extracted and stored at −20 °C until analysis.

#### 2.1.2. Questionnaires and Measurements

Information about education level and smoking habits was collected from questionnaires. Height and weight were measured in light clothing and without footwear.

To collect dietary data, an extensive previously validated FFQ [24], developed at the University of Oslo (UiO), was used to assess food and nutrient intake during the last year. The questionnaire included measures on frequency and amount of food intake, in addition to open questions. The FFQ was handed out to all Tromsø 7 participants at the examination site and could be completed at the examination site or at home and returned by mail with a prepaid envelope. Estimations of intakes of energy, food, and macro- and micronutrients intake was performed at UiO using the food composition database software system Kostberegningssystemet (KBS, version 7.3), database version AE14, based on the Norwegian Food Composition Tables from 2014 and 2015 [25]. The estimations of iodine intake were done in the AE18 food composition database. Detailed descriptions of the Tromsø 7 FFQ data collection and processing are presented elsewhere [26]. 

The FFQ was designed to assess habitual diet the preceding year. Frequencies of intake ranged from never, per month, per week, to several times a day, varying with the type of food item in question. Amounts were given in household measures, units of liter, deciliter, a piece, a slice, etc., or as portion sizes. Questions about supplement use were included. 

Among the 493 participants included in the urine analysis (see above), 450 participants completed the FFQ and were included in this analysis. 

### 2.2. Urine Analysis

Urine iodine analyses were done at the Hormone Laboratory, Oslo University Hospital, Norway. The UIC was measured colorimetrically using the Sandell–Kolthoff’s reaction based on the catalytic effect of iodine on the redox reaction between arsenic and cerium after ammonium persulfate digestion of the samples. Intra- and inter-assay coefficients of variation (CVs) were 8%. The Hormone Laboratory is accredited as a testing laboratory by Norwegian Accreditation according to the standard NS-EN ISO/IEC 17025, with Registration number TEST 099.

Daily urinary iodine excretion (UIE) was calculated using the following formula:(1)24-h UIE (µg/day) = UIC (µg/L) × 24-h urine volume (L).

Daily iodine intake (g/day) was estimated using the following formula [20]: (2)Daily 24-h UIE (µg/day)/0.83.

This is based on an estimate of 92% bioavailability and 90% excretion of iodine in an ordinary diet where iodine is in the form of salt and also protein bound. 

The median UIC of the participants was compared with the iodine status criteria developed by the WHO, UNICEF, and International Council for the Control of Iodine Deficiency Disorders (ICCIDD) for adults [1].

### 2.3. Ethics

The study adhered to the tenets of the Declaration of Helsinki and the Norwegian Data Protection Authority. The Regional Committee of Medical and Health Research Ethics North approved Tromsø 7 (REK 2014/940). This substudy was approved by the Regional Committee of Medical and Health Research Ethics (REK 2016/1795). All participants gave written informed consent.

### 2.4. Statistical Analysis

Analysis of the data was performed using IBM SPSS statistical software (V.22 SPSS Inc., Chicago, IL, USA). Descriptive statistics are presented as mean with standard deviation (SD) and as medians, with 25th and 75th percentiles for variables that were not normally distributed. To compare variables, relevant statistics such as independent samples *t*-test, Mann–Whitney U Test, and Spearman correlation analysis and regression models were performed. *p*-values < 0.05 were considered to be statistically significant.

For the comparisons of iodine intake estimated from 24-h urine and FFQ, we used Spearman’s correlation coefficient. Bland–Altman plots with limits of agreement were constructed to evaluate whether the difference between estimated iodine intake from 24-h urine and FFQ varied across the mean iodine intake from the two methods and the agreement between the two methods. Two influential points were excluded when testing correlations (extreme outliers that completely changed the direction of the line and gave inflated limits of agreement). We performed additional analysis excluding those who reported that their urine collection was incomplete, completed less than 90% of the FFQ, and participants having very high or low total energy intake (the 1% highest and 1% lowest), in total 190 participants, leaving 303 participants for sensitivity analysis. 

## 3. Results

### 3.1. Participant Characteristics

The study sample consisted of 493 participants (51% women) with a mean age of 56 years, mean BMI around 27 kg/m^2^, and more than 50% had tertiary education (Table 1). 

The total mean 24-h creatinine concentration was 9.7 (SD 4.1) mmol/L; men had higher creatinine value than women (11.7 (SD 5.4) and 7.9 (SD 4.6), respectively) (*p* < 0.01).

### 3.2. Iodine Excretion and Estimated Iodine Intake

The median (25th and 75th percentiles) 24-h urine volume for the study population was 1.69 L (1.3, 2.20) and was not statistically different between women and men (*p* = 0.06) (Table 2). The total 24-h median UIC (25th and 75th percentiles) in all participants was 88 µg/L (38, 100), and 63 and 100 µg/L in women and men, respectively, (*p* = 0.14) (Table 2). Around 16% of the participants (23 and 8% of women and men, respectively) had UIC values below 50 µg/L, while 1.4% had UIC > 600 µg/L. 

The median (25th and 75th percentiles) UIE for the study population was 132 (87, 200) µg/day. Men had higher UIE values than women (*p* < 0.01). 

The median daily iodine intake for the study population estimated from UIE (on the basis that 83% of ingested iodine is excreted) was 158.9 µg/day (174 in men and 133 µg/day in women, *p* < 0.001) and did not differ significantly with respect to BMI or education levels. 

Median iodine intake (estimated from UIE) was significantly higher in men aged 55–69 years as compared with men aged 40–54 years (*p* < 0.001), but no significant age difference was found in women (*p* = 0.14).

### 3.3. Estimated Iodine Intake from Diet and Supplements (FFQ)

Overall, the estimated daily iodine intake of the 450 participants who completed the FFQ ranged from 38 to 1165 µg/day with a median (25th, 75th percentiles) iodine intake of 281 µg/day (212.5, 378.5). Men had significantly higher iodine intake than women (*p* < 0.001) (Table 3). The iodine intake was still higher among men after adjusting for energy intake (data not shown).

In women, 10.7% had iodine intake below the recommended daily intake (150 µg/day), while only 4% had intake below the EAR (100 µg/day) and the proportion of upper intake levels (UL > 600) of iodine intake was 3%. 

In men, 7.9% had iodine intake below the recommended daily, almost no men had intake below the EAR intake, while 7.8% had excessive intake.

The estimated iodine intake was significantly higher in the older age group, both in men (*p* = 0.01)) and in women (*p* = 0.02). Iodine intake did not differ significantly with respect to education levels or BMI in men and women.

### 3.4. Sources of Iodine

Table 4 shows intake of typical iodine source food groups in the study population.

The intake of dairy products and fish was higher among men than women (*p* < 0.001), but there were no differences in egg consumption. 

In women, there was a moderate correlation between total iodine intake estimated from diet and intake of milk and milk products (Spearman rank correlation coefficient, r = 0.49, *p* < 0.01), and the Spearman rank correlation coefficient between iodine intake and fish and fish product, and eggs was r = 0.65 (*p* < 0.01) and r = 0.23 (*p* < 0.01), respectively. 

In men, the correlation between total iodine intake estimated from diet and intake of milk and milk products was r = 0.47 (*p* < 0.01), intake of fish showed a correlation of r = 0.68 (*p* < 0.01), and intake of eggs showed a correlation of r = 0.17 (*p* = 0.01).

### 3.5. Supplements

Around 13% (*n =* 64) of the participants reported taking multivitamin and mineral supplements. Among these, 61% (*n =* 39) took the supplements daily. The estimated median iodine intake from diet was 448 µg/day among daily users of dietary supplements, and significantly different (*p* < 0.001) from the median 270 µg/day among no supplement users (*n* = 396). 

The median intake of iodine in female (*n* = 24) and male (*n* = 15) daily supplement users was 415 and 488 µg/day, respectively. The median iodine intake was higher in the older age group (490 and 383 µg/day, respectively).

### 3.6. Relationship between the Two Methods for Estimating Iodine Intake

Among the 450 participants who completed the FFQ and provided a 24-h urine sample, the median iodine intake calculated from the FFQ was significantly higher than the iodine intake estimated from the UIE (*p* < 0.001). 

In all participants combined, there was a moderate correlation between total iodine intake estimated from the FFQ and the UIE (Spearman rank correlation coefficient, r = 0.39, *p* < 0.01). The correlation was r = 0.30 (*p* < 0.01) and r = 0.39 (*p* < 0.01) in women and men, respectively. 

The differences between paired estimates from UIE and FFQ are plotted against the mean values of the paired estimates (Bland–Altman plot).

The Bland–Altman plots showed that the FFQ overestimated the intake of iodine as compared with the UIE, and that the differences between the methods increased with increasing intakes. The mean differences were 119 (SD 162) µg/day in women and 87 (SD 280) µg/day in men (Figure 1A,B).

### 3.7. Sensitivity Analyses

In subanalysis, excluding those who reported that their urine collection was incomplete or completed less than 90% of the FFQ, and participants having very high or low total energy intake (the 1% highest and 1% lowest), we found that the estimated intake of iodine, calculated from 24-h urine excretion or the FFQ, was not significantly different from the whole sample (before exclusion). For women, the median daily iodine intake calculated from the UIE was 133.2 and 138.4 μg/day before and after exclusion, respectively. The corresponding figures in men were 174.4 versus 179.5 μg/day. Furthermore, the median daily iodine intake estimated from the FFQ was 263 and 262 µg/day in women and 317.5 and 305 µg/day in men, before and after exclusion, respectively. The median energy intake for women and men was 8.6 and 10.7 MJ/day before exclusion (*n* = 450) and 8.9, and 10.4 MJ/day after exclusion (*n* = 303).

## 4. Discussion

In the present study, iodine intake was assessed both by 24-h UIE and by using a FFQ in an adult population aged 40 to 69 years living in Tromsø, Norway. The median iodine intake estimated from 24-h UIE was 159 µg/day (174 and 133 µg/day in men and women, respectively) and, from the FFQ, the median iodine intake was 281 µg/day (317 and 263 µg/day in men and women, respectively). The iodine intake estimated from 24-h UIC describes a mildly iodine deficient female population, while the iodine intake findings based on these two methods indicate that the adult male study population have iodine intake within recommended values. 

Inadequate iodine status has been documented in different groups in Norway and in other Nordic countries, mainly pregnant and lactating women [5,15,16,17], but studies in a representative adult population is still lacking in Norway. However, our findings are in line with what has been found in a recent study on iodine intake in Norway with a convenience sample [9]. 

### 4.1. Variation in Iodine Intake by Gender, Age, and Education

The median energy intakes in women and men of 8.6 and 10.7 MJ/day, respectively, are similar to what was observed previously in other Norwegian population-based studies [12,27]. 

An average higher iodine intake is expected in men as compared with women, because men have higher food and energy intake. In our study, men had 25% higher energy intake and around 20% higher iodine intake as compared with women, even when adjusting for energy intake. These different intakes between men and women are in agreement with earlier studies that showed a low iodine intake among women in Norway [12] and in Switzerland [28]. 

Compared with the findings from other Norwegian dietary surveys among adult men and women, our population had a relatively higher median intake of iodine [12,29]

Iodine estimated from the FFQ and UIE was significantly higher in men aged 55–69 as compared with those 49–54 years, but this age difference was not found in women. Although a lower intake of iodine among younger women has previously been reported in Norway [29], the reason for our findings is not clear. One explanation can be the small spread in age in this group. The same tendency of age and gender difference in iodine intake, although not significant, have been documented in the Danish population [30]. Furthermore, as previously documented, the iodine intake was not related to education level [31,32].

### 4.2. Iodine Intake as Compared with Recommendations

To our knowledge, this is the first population-based study among healthy adults in Norway to evaluate how iodine intake estimated from FFQ compares with iodine intake estimated by 24-h urinary iodine excretion. 

According to the WHO criteria (assuming that a single 24-h UIC value in reality is the mean of a number of spot urine samples from an individual), iodine deficiency is defined as a population median UIC below 100 μg/L. The UIC, in this study, was 63 and 100 µg/L in women and men, respectively, indicating that iodine nutrition is not likely to be a problem for men but denotes an iodine deficiency among women. Apparently, the proportion of UIC values <50 μg/L was 23% for women, which is above the recommendation that no more than 20% of the participants should be below 50 μg/L. Even when adjusted for the potential dilution effect from large urine volumes of about 1.8 L in women to the normalized mean volume of 1.5 L, the median UIC (76 µg/L) is still below the current WHO criteria [33].

The median UIC among men in this study was higher as compared with Somali immigrant men living in Oslo, who had a 24-h UIC of 63 μg/L, but similar in women [31]. The 24-h UIC in our population was closer to that reported for adults in Norway [9] and in Switzerland [28]. 

The distribution of intake, calculated from FFQ, was also assessed to determine the proportion of the population below estimated average requirements (EAR) using the EAR cut-point approximation [34]. Around 4% of women and almost none of men had intakes below the EAR of 100 µg/day, indicating adequate iodine intake in men, and very close to adequate intake in women; expecting 2.5% of the population below the cut-off at sufficiency.

Although the results showed a correlation between the two methods, the estimated iodine intake calculated from FFQ was much higher as compared with iodine estimated from UIE. The FFQ dietary assessment of iodine intake aims to measure the average long-term iodine intake from the total habitual diet. Dietary assessment using FFQs largely depends on the participants memory and perceptions [35]. As with all self-reporting dietary data, the present dietary assessment can be affected by an external bias caused by social desirability and memory lapses [36]. The findings also illustrate that the two methods cannot be used interchangeably (Figure 1).

The higher iodine intake, estimated from FFQ analysis in the present study, may at least partly be explained by an over-reporting of healthy food, such as fish intake. The iodine content of the main dietary sources in Norway is well characterized, but for other food groups the food composition data may introduce more uncertainty. In addition, the iodine content in otherwise comparable foods may vary, due to natural variation in soil, water, fertilizers and the use of iodine enrichment. Similar discrepancies of iodine intake estimated from UIC and FFQ methods have been reported [30,37,38].

The large spread in the individual urine volumes in this study is supported by data from other studies [39,40]. Although the participants were informed not to change their daily routines and habits during the urine collection, we are not sure whether our findings represent the habitual urinary volume excretion, or whether participants changed their habits and routines, for example stayed home the day they were collecting urine and therefore drank more fluids. However, excluding persons who stated that they did not collect urine samples fully, hardly affected our results. 

### 4.3. Study Limitations and Strengths

The main strength of this study was the randomly selected sample from a population-based study with high participation, the iodine intake estimated from both FFQ and 24-h urinary iodine excretion, and urine samples being analyzed in one batch. Twenty-four-hour urinary iodine excretion is the most reliable biochemical marker for assessing iodine status. Other strengths in this study was that the data collection was conducted during nine months (August 2015–April 2016), although it did not include the summer months when milk and milk products have a lower iodine concentration than during winter months due to increased use of fortified cattle feed during the winter period [41].

The main limitation of this study is due to day-to-day variation, because the 24-h urine was collected only once, whereas multiple measurements is optimal. Another limitation is the risk of selection bias common in population-based studies, which cannot be ruled out. In addition, the attendance was rather high in the Tromsø Study as compared to many other population-based studies. A comparison between responders and non-responders randomly invited for this substudy showed no statistically significant differences in education levels or BMI. Finally, the FFQ data are self-reported, and thus there is a risk of bias resulting in over-reporting of healthy food such as fish. The other limitation is that the study does not include adults in the age group 18–39 years.

## 5. Conclusions

The current study on a healthy adult population in Norway shows that iodine intake estimated from 24-h UIC and its derivative measures, describes a mildly iodine deficient female population, while the male population is iodine sufficient. Concurrent use of an extensive FFQ describes both sexes as iodine sufficient.

Further studies, applying a dietary assessment method validated for estimating iodine intake and repeated individual urine collections, are required to determine the habitual iodine intake in this population.

## Figures and Tables

**Figure 1 nutrients-12-03246-f001:**
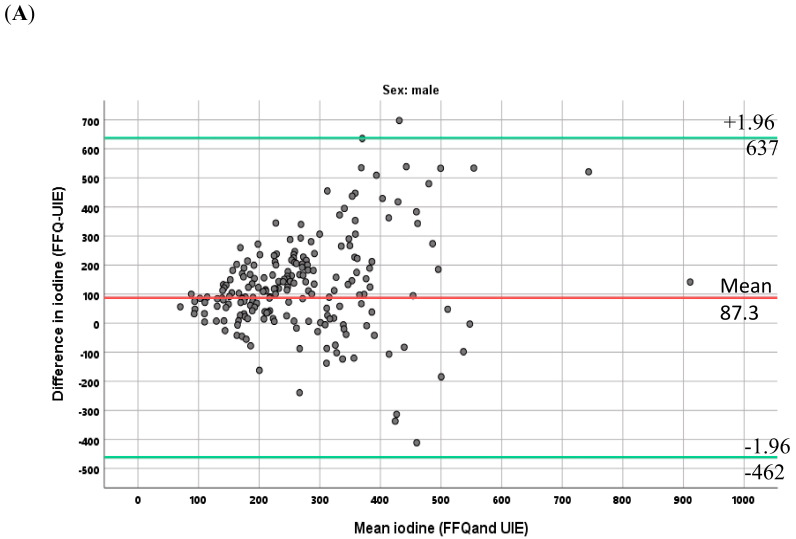
Bland–Altman plot, mean iodine intake estimated from 24 h-urine and the FFQ versus the difference between the methods (FFQ-UIE), in men (**A**) and women (**B**), The vertical line indicates mean difference in iodine intake estimates (red line), with the representation of the limits of agreement (green lines), from −1.96 SD to +1.96 SD.

**Table 1 nutrients-12-03246-t001:** Characteristics of the women and men who collected 24-h urine in the Tromsø Study 2015–2016 (*n* = 493).

	All	Women	Men
Gender, *n*	493	252	241
Age, years	56 (8.4)	57 (8.4)	55 (8.4)
Primary education (up to 10 yrs.), %	20.4 (100)	21.5 (54)	19.2 (46)
Secondary education (up to 13 yrs.), %	28.3 (139)	27.1 (68)	29.6 (71)
Tertiary education (university), %	51.3 (252)	51.4 (129)	51.2 (123)
Body mass index, kg/m^2^	27.2 (4.3)	27.5 (3.7)	26.8 (4.7)

Values are mean (standard deviations) or percentages (numbers). yrs, years.

**Table 2 nutrients-12-03246-t002:** Median and interquartile ranges of 24-h urine volume, 24-h urine iodine concentration (UIC), 24-h iodine excretion (UIE), and 24-h iodine intake by sex and age stratification in men and women aged 40–69 years in the Tromsø Study 2015–2016.

Total Population		Women (*n* = 252)			Men (*n* = 241)		
	All (40–69 Years (*n* = 493)	All	40–54 Years (*n* = 112)	55–69 Years	All	40–54 Years	55–69 Years
		(*n* = 252)		(*n* = 140)	(*n* = 241)	(*n* = 90)	(*n* = 151)
24-h urine volume (L)							
Mean (SD)	1.74 (0.59)	1.77 (0.60)	1.74 (0.62)	1.79 (0.58)	1.67 (0.57)	1.64 (0.61)	1.70 (0.55)
Median (25th, 75th) *	1.69 (1.31, 2.20)	1.80 (1.29, 2.30)	1.73 (1.24, 2.35)	1.80 (1.36, 2.32)	1.62 (1.27, 2.10)	1.53 (1.13, 2.21)	1.65 (1.32, 2.01)
24-h UIC (µg/L)							
Median (25th, 75th)	88 (50, 125)	63 (50, 100)	63 (38, 100)	75 (50, 113)	100 (63, 150)	88 (63, 125)	100 (63, 153)
24-h UIE (μg/day) ^#^							
Mean (SD)	199 (545)	182 (724)	222 (1079)	150 (116)	214 (247)	150 (96)	253 (297)
Median (25th, 75th) *	132 (87, 200)	111 (78, 170)	102 (79, 149)	117 (75, 184)	145 (108, 234)	132 (95, 176)	168 (113, 265)
24-h iodine intake (μg/day) ^†^							
Mean (SD)	241.6 (670)	222.5 (890)	271.6 (1327)	183.2 (141)	261.6 (302)	182.9 (117)	308.5 (364)
Median (25th, 75th) *	158.9 (106, 245)	133.2 (93, 205)	122.8 (95, 182)	141.1 (91, 223)	174.4 (129, 284)	159.3 (114, 216)	202.9 (135, 321)

* 25th percentile and 75 percentile, ^#^ 24-h iodine excretion (µg/day) = UIC (µg/L) × 24-h urine volume (L/day) and ^†^ 24-h iodine intake, estimated from 24-h UIE divided by intake/excretion ratio of 0 83.

**Table 3 nutrients-12-03246-t003:** Estimated daily habitual iodine intake (µg/day) from food frequency questionnaire in men and women aged 46–69 years, Tromsø Study 2015–2016.

Subgroup	*n*	Mean (SD)	Median (P25, P75) *
All (40–69 years)	450	314.3 (157.1)	281 (212.5, 378.5)
Women	234	288.1 (142.9)	263 (200, 355.3)
40–54 years	102	271.6 (151.9)	244 (186.5, 328.3)
55–69 years	132	300.8 (142.9)	278.5 (216.3, 371.3)
Men	216	342.7 (166.7)	317.5 (228.3, 407.5)
40–54 years	80	304.6 (133)	295 (215.8, 367.8)
55–69 years	136	365.2 (189.6)	329 (246.3, 447.3)

* 25th and 75th percentile.

**Table 4 nutrients-12-03246-t004:** Estimated mean intake of dairy products, fish, and eggs among men and women aged 46–69 years, Tromsø Study 2015–2016.

	*N*	Dairy (g) (Min–Max)	Fish (g) (Min–Max)	Eggs (g) (Min–Max)
All (40–69 yrs)	450	496.6 (0–3061)	119.3 (0–606)	27.1 (0–234)
Women	234	422.6 (0–1957)	108.2 (0–606)	28.3 (0–234)
40–54 (yrs)	102	412.8 (0–1957)	92.1 (0–606)	32.9 (0–234)
55–69 (yrs)	132	430.1 (30–1657)	120.6 (0–375)	24.7 (0–233)
Men	216	576.0 (16–3016)	131.2 (0–489)	25.8 (0–186)
40–54 (yrs)	80	596.7 (16–3061)	107.9 (0–360)	28.0 (0–101)
55–69 (yrs)	136	563.8 (26–1919)	144.9 (10–489)	24.4 (0–186)

Min, Minimum; Max, maxium. yrs, years.

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
