# Peer review of "Iodine Intake in Norwegian Women and Men: The Population-Based Tromsø Study 2015–2016"

_nutrients, 2020, doi:10.3390/nu12113246_

Round 1
Reviewer 1 Report
The manuscript by Madar et al is a profound study the current iodine status and intake in a Norwegian adult population. The manuscript is well written and data is good presented. The definite strength of the manuscript is the use of two independent methods: iodine measurements in 24-hour urine samples and a food frequency questionnaire. Methods are described in detail. Results are very well discussed including strength and limitations.
Minor comments to the authors: data presentation in Table 2 is shifted (24-hour iodine intake), and it is "Men" and not "Menn" in subheading. There is an indicated number for Men (N=241), but not for Women (?). I also suggest authors to highlight the most interesting data in the Table 2 with bold font or colour.
Author Response
- Minor comments to the authors: data presentation in Table 2 is shifted (24-hour iodine intake), and it is "Men" and not "Menn" in subheading. There is an indicated number for Men (N=241), but not for Women (?). I also suggest authors to highlight the most interesting data in the Table 2 with bold font or colour.
Reply: The has been done and the table is updated. Some of the data in table 2 is bolded but we are not sure the policy of Nutrients regarding bolding the most important data.
Reviewer 2 Report
The paper is well written. Some minor points are listed below.
- P2 line 66
“ ….and a median UIC higher than 600 µg/L is defined as ‘excessive iodine intake’(21)”
By WHO (World Health Organization/United Nations Children’s Fund/International Council for Control of Iodine Deficiency Disorders. Assessment of the iodine deficiency disorders and monitoring their elimination: a guide for programme managers. – 3rd ed. Geneva: WHO, 2007) or UNICEF (Guidance on the Monitoring of Salt Iodization Programmes and Determination of Population Iodine Status, 2019, https://www.ign.org/guidance-on-the-monitoring-of-salt-iodization-programmes-and-determination-of-population-iodine-status.htm), an excessive iodine intake of a given population is defined as a median UIC of > 300 µg/L. However, it seems the definition of excess iodine intake is a median UI higher than 600 µg/L in Norway. Is there special reason that the definition of excessive iodine intake is higher in Norway?
- Table 2
There is a spelling error of “Menn”.
- P3 line 250-252
“The median iodine intake……. µg/day respectively” is a repeated sentence.
Author Response
- P2 line 66 “ ….and a median UIC higher than 600 µg/L is defined as ‘excessive iodine intake’(21)”
By WHO (World Health Organization/United Nations Children’s Fund/International Council for Control of Iodine Deficiency Disorders. Assessment of the iodine deficiency disorders and monitoring their elimination: a guide for programme managers. – 3rd ed. Geneva: WHO, 2007) or UNICEF (Guidance on the Monitoring of Salt Iodization Programmes and Determination of Population Iodine Status, 2019, https://www.ign.org/guidance-on-the-monitoring-of-salt-iodization-programmes-and-determination-of-population-iodine-status.htm), an excessive iodine intake of a given population is defined as a median UIC of > 300 µg/L. However, it seems the definition of excess iodine intake is a median UI higher than 600 µg/L in Norway. Is there special reason that the definition of excessive iodine intake is higher in Norway?
Reply: The safe upper level (UL) for adults of 600 µg/d is proposed in the Nordic Nutrition Recommendations (NNR 2012), reference number 21 in our manuscript. It is based on proposed safe upper level (UL) of iodine intake by the Scientific Committee for Food (ref: Opinion of the Scientific Committee on Food on the tolerable upper intake level of iodine (expressed on 26 September 2002): Scientific Committee on Food 2002). However, the NNR committee also included that they did not rule out the possibility of adverse effects of a UIC above 300µg/L.
- Table 2
There is a spelling error of “Menn”.
Reply: This has been done.
- P3 line 250-252
“The median iodine intake……. µg/day respectively” is a repeated sentence.
Reply: The sentence has been updated now.